# Comparative Transcriptome and Anatomic Characteristics of Stems in Two Alfalfa Genotypes

**DOI:** 10.3390/plants11192601

**Published:** 2022-10-03

**Authors:** Jierui Wu, Xiaoyu Wang, Ruxue Zhang, Qingwen Fu, Fang Tang, Fengling Shi, Buhe Temuer, Zhiqiang Zhang

**Affiliations:** 1Key Laboratory of Grassland Resources of the Ministry of Education, Technology Engineering Center of Drought and Cold-Resistant Grass Breeding in the North of the National Forestry and Grassland Administration, College of Grassland, Resources and Environment, Inner Mongolia Agricultural University, Hohhot 010010, China; 2Key Laboratory of Forage Cultivation, Processing and High Efficient Utilization of Ministry of Agriculture, Inner Mongolia Agricultural University, Hohhot 010010, China

**Keywords:** *Medicago sativa* L., stems, lignin, transcriptome, anatomic

## Abstract

Stems are more important to forage quality than leaves in alfalfa. To understand lignin formation at different stages in alfalfa, lignin distribution, anatomical characteristics and transcriptome profile were employed using two alfalfa cultivars. The results showed that the in vitro true digestibility (IVTD) of stems in WL168 was significantly higher than that of Zhungeer, along with the significantly lower neutral detergent fiber (NDF), acid detergent fiber (ADF) and lignin contents. In addition, Zhungeer exhibited increased staining of the xylem areas in the stems of different developmental stages compared to WL168. Interestingly, the stems of WL168 appeared intracellular space from the stage 3, while Zhungeer did not. The comparative transcriptome analysis showed that a total of 1993 genes were differentially expressed in the stem between the cultivars, with a higher number of expressed genes in the stage 4. Of the differentially expressed genes, starch and sucrose metabolism as well as phenylpropanoid biosynthesis pathways were the most significantly enriched pathways. Furthermore, expression of genes involved in lignin biosynthesis such as *PAL*, *4CL*, *HCT*, *CAD*, *COMT* and *POD* coincides with the anatomic characteristics and lignin accumulation. These results may help elucidate the regulatory mechanisms of lignin biosynthesis and improve forage quality in alfalfa.

## 1. Introduction

Alfalfa (*Medicago sativa* L.) is a high-quality legume forage with strong adaptability and high yield. In China, alfalfa planting area has expanded rapidly, and the demand for high-quality alfalfa is also increasing. Stems and leaves are the mainly feed parts in alfalfa, but the proportion of stems is generally in 50–70% [1]. Alfalfa stems are mainly composed of cell walls and cell content, of which more than 80% is cell walls [2]. The cell wall composition of secondary xylem of alfalfa stem is approximately 400 g·kg^−1^ cellulose, 200 g·kg^−1^ hemicellulose, 200 g·kg^−1^ pectin and 200 g·kg^−1^ lignin [3]. The hierarchical structure and condensed structure of cellulose itself in the cell wall, as well as the lignin that wraps cellulose, make it difficult to digest in animal rumen, and also reduces the digestibility of other organic matter [4]. Studies have shown that the content and composition of lignin directly affect the quality [5,6], utilization and lodging resistance of alfalfa [7]. Only the green tissue of alfalfa is never lignified, and there is much more lignified tissue in the stem is than that in leaf [3]. Therefore, the structural characteristics of stems have a greater impact on the forage quality and utilization compared with those of leaves in alfalfa. Many studies have focused on the anatomical structure of the stem and the lodging resistance of crops [8,9]. It was found that the stem mechanical strength was positively correlated with lignin accumulation in arabidopsis thaliana [10]. In addition, the plant density, light intensity and fertilization level also affect plant stem development [11,12]. Lignin content varies with crop or forage cultivars, resulting in different lodging resistance or forage digestibility.

The components of cell walls are closely related to the cultivar and developmental stage in alfalfa [2]. There are detailed studies on the anatomy of alfalfa stems. The cambium, phloem and primary xylem parenchyma of alfalfa stems all have very thin primary and secondary wall structures, and lignification first starts from cells near the xylem, and gradually expands to the middle [1]. Lignin is predominantly deposited in secondarily thickened cell walls of vascular plants and plays important roles in cell wall structural integrity, stem strength, water transport, mechanical support and plant pathogen defense [13,14], but the components and contents of lignin hinder the digestion and degradation of plant cell walls [15]. Due to the adverse effects of lignin in stem on the digestibility and biomass conversion efficiency of alfalfa during feeding, there have been several research hotspots in recent years, including reducing lignin content or changing lignin composition to increase its degradation, and improving the feed value, energy development and utilization efficiency of alfalfa [3,7]. The reduced lignin alfalfa cultivar HarvXtra was achieved by downregulating the gene encoding caffeoyl-CoAO- methyltransferase [16]. HarvXtra-008 is lower in acid detergent lignin (ADL) by 7 to 10% and amylase-treated neutral detergent fiber by 2 to 10%, and its has 4 to 9% greater neutral detergent fiber digestibility (NDFD) than non-reduced lignin cultivars [17].

Lignin is a polymer composed of three kinds of lignin monomer: p-hydroxyphenyl lignin (H), guaiacyl lignin (G) and syringyl lignin (S) [18]. Numerous studies have identified the enzymes and genes responsible for lignin biosynthesis, including phenylalanine ammonia-lyase (*PAL*), cinnamate 4-hydroxylase (*C4H*), 4-coumarate-CoA ligase (*4CL*), shikimate/quinate hydroxycinnamoyl transferase (*HCT*), coumarate3-hydroxylase (*C3H*), caffeoyl shikimate esterase (*CSE*), ferulate5-hydroxylase (*F5H*), caffeic acid 3-O-methyltransferase (*COMT*), caffeoyl-CoAO-methyltransferase (*CCoAOMT*), cinnamoyl-CoA reductase (*CCR*), cinnamyl alcohol dehydrogenase (*CAD*), peroxidase (*POD*) and laccase (*LAC*) [19,20]. Conventional breeding based on phenotypic selection for quality improvement in alfalfa is time consuming. In order to decrease lignin content in alfalfa, most studies have focused on down-regulating key genes in the lignin biosynthesis pathway, which significantly increases forage digestibility [21,22,23]. These results indicate that both lignin content and composition could affect forage digestibility in alfalfa. However, genetic modifications in lignin biosynthesis are often associated with undesirable traits, such as dwarf phenotypes, resulting in huge biomass loss [24].

To understand lignin formation at different stages of different alfalfa cultivars, lignin distribution, anatomical characteristics and transcriptome profile were employed using two alfalfa cultivars, Zhungeer and WL168, with different stem degradability. The anatomical structures, lignin content and transcriptome profiles of both alfalfa cultivars during the four developments of stems were observed and analyzed. Our results may provide new ideas and theoretical bases for regulatory mechanisms of lignin biosynthesis and quality improvement in breeding alfalfa.

## 2. Results

### 2.1. Nutritional Quality Analysis

In order to evaluate the forage quality of the stem in both cultivars, IVTD, NDF, ADF and hemicellulose, which are important indices for forage quality, were determined. As shown in Figure 1, the IVTD of stems in WL168 was significantly higher than in Zhungeer, along with significantly lower NDF and ADF and higher hemicellulose contents. In detail, WL168 showed a 4.54% increase in stem IVTD, a 2.97% decrease in stem NDF, a 7.62% decrease in stem ADF and a 4.63% increase in stem hemicellulose compared to Zhungeer. 

### 2.2. Lignin Content Analysis

To investigate how lignin changed with development stage between two cultivars, the lignin content of the stem in both cultivars was measured by the acetyl bromide procedure [25]. As shown in Figure 2, lignin content of stems in both cultivars increased gradually with the development of maturity, but the lignin content of stems in Zhungeer was significantly higher than in WL168 in the all development stages. In detail, the total lignin content from stage 1 to stage 4 in Zhungeer was 19.46, 21.48, 24.99 and 28.47% higher than those of WL168, respectively.

### 2.3. Stem Structure Analysis

In terms of anatomic characteristics, the xylem areas in the stems of both cultivars thickened and expanded as the stems developed, while Zhungeer exhibited clearer and stronger lignin staining in the phloem fiber in the stems at different developmental stages compared to WL168 (Figure 3). In addition, intracellular space appeared in the stems of WL168 from the third stage (H3,4), but not in Zhungeer stems (S3,4) (Figure 3).

### 2.4. Summary of Transcriptome Sequencing, Assembly and Annotation

Following clean-up and quality filtering, 78.21 Gb clean reads from WL168 and 78.65 Gb from Zhungeer were obtained, respectively. The clean data of each sample reached more than 5.71 Gb. A transcriptome datebase containing 125,093 transcripts of average length 1148.69 bp was obtained, with an N50 length of 1668 bp. Among these unigenes, 39,543 (70.54%) were longer than 500 bp (Table 1).

All unigenes and transcripts obtained by transcriptome assembly were aligned with eight major databases (COG, GO, KEGG, KOG, Pfam, Swiss-prot, eggNOG and Nr databases). A total of 56,054 unigenes were annotated in the eight databases. Of the 56,054 assembled unigenes, 45,398 were found to have homologs in the databases COG (15,281 unigenes, 27.26%), GO (25,913 unigenes, 46.23%), KEGG (17,045 unigenes, 30.41%), KOG (23,860 unigenes, 42.57%), Pfam (30,672 unigenes, 54.72%), Swiss-prot (25,608 unigenes, 45.68%), eggNOG (40,075 unigenes, 71.49%) and Nr (44,610 unigenes, 79.58%) (Table 2). 

### 2.5. Differentially Expressed Genes and Functional Enrichment Analysis

The differentially expressed genes between both the cultivars among the four developmental stages were analyzed. A total of 1993 significant differentially expressed genes were detected. In the first stage, 494 unigenes were identified as significant differentially expressed genes, which included 188 up-regulated and 306 down-regulated unigenes (H1 vs. S1). In the second stage, a total of 268 up-regulated and 175 down-regulated unigenes were identified (H2 vs. S2). Only 62 up-regulated and 11 down-regulated unigenes were identified in the third stage (H3 vs. S3), while 487 up-regulated and 819 down-regulated unigenes were observed between the two cultivars in the fourth stage (H4 vs. S4) (Figure 4). 

To further understand the function of differentially expressed genes, functional annotations of the differential expressed genes were conducted by comparing the unigene sequences with the GO databases. The GO terms for the differentially expressed genes were classified into three main classes: biological process, cellular component and molecular function. Within the category of biological processes, in H1 vs. S1 most of the differentially expressed genes were assigned to metabolic process (137 unigenes, 27.73%), cellular process (115 unigenes, 23.28%) and single-organism process (101 unigenes, 20.45%). In the cellular component category, most differentially expressed genes were distributed to membrane (101 unigenes, 20.45%), membrane part (86 unigenes, 17.41%), cell (71 unigenes, 14.37%) and cell part (71 unigenes, 14.37%). Catalytic activity (142 unigenes, 28.74%) and binding (110 unigenes, 22.27%) were the most highly represented groups under the molecular function category. Annotation for genes differentially expressed in H2 vs. S2, H3 vs. S3 and H4 vs. S4 pairwise comparisons was also carried out. In all three functional categories, genes belonging to the above three groupings showed similar distribution patterns to H1 vs. S1 (Table 3).

Furthermore, a gene set enrichment analysis of the differentially expressed genes was performed for the KEGG annotations to determine over-represented functional pathways for different genotypes and development stages. In the first stage (H1 vs. S1), 134 differential expressed genes were enriched into 63 pathways, of which the starch and sucrose metabolism (11 unigenes), phenylpropanoid biosynthesis (10 unigenes), isoflavonoid biosynthesis (6 unigenes) and flavonoid biosynthesis (5 unigenes) were significantly enriched (with a *p*-value < 0.05). There were 107 differential expressed genes enriched into 58 pathways in the second stage (H2 vs. S2), among which the starch and sucrose metabolism (7 unigenes), phenylpropanoid biosynthesis (5 unigenes) and plant-pathogen interaction (5 unigenes) were the most enriched pathways, while only 10 differential expressed genes were enriched into 9 pathways in stage 3 (H3 vs. S3). In the fourth stage (H4 vs. S4), 451 differential expressed genes were grouped into 105 pathways, among which the plant-pathogen interaction (15 unigenes), carbon fixation in photosynthetic organisms (10 unigenes), fatty acid elongation (6 unigenes), fatty acid biosynthesis (5 unigenes) and tyrosine metabolism (5 unigenes) were the most enriched pathways. It is worth noting that there are four significant enrichment pathways in the first stage (H1 vs. S1), namely, starch and sucrose metabolism (ko00500), phenylpropanoid biosynthesis (ko00940), isoflavone biosynthesis (ko00943) and flavonoid biosynthesis (ko00941). There are no significant enrichment pathways in the other three groups (Figure 5).

### 2.6. Expression Profiles of Candidate Genes Involved in Lignin Biosynthesis at Different Developmental Stages in Alfalfa

In order to identify key genes responsible for differences in stem properties between the cultivars at different developmental stages, the relative transcript levels of 12 genes (*PAL* (1 unigene), *4CL* (1 unigene), *HCT* (2 unigenes), *CAD* (2 unigenes), *COMT* (3 unigenes) and *POD* (3 unigenes) encoding the key enzymes related to lignin biosynthesis (phenylpropanoid) pathway were analyzed. The results showed that the FPKM (fragments per kilobase of exon model per million mapped fragments) of 8 genes encoding *PAL*, *HCT*, *COMT* and *POD* showed an upward trend with the increase of alfalfa maturity in both the alfalfa cultivars, but 1 gene encoding *4CL* and 1 gene encoding *HCT* showed an upward trend in Zhungeer and a downward trend in WL168, respectively. Moreover, one gene encoding *CAD* showed an upward trend in WL168 and a downward trend in Zhungeer, but the other gene encoding *CAD* showed a downward trend in both the cultivars. The expression levels of 10 genes, including *PAL*, *4CL*, *HCT*, *COMT* and *POD*, involved lignin biosynthesis in Zhungeer were higher than those in WL168 except for 2 genes encoding *CAD*, which is consistent with the change trend of lignin content (Figure 6).

## 3. Discussion

The decline of alfalfa quality with maturity can be partially attributed to a decrease in leaf and increase in stem proportion as the plant matures [26]. Because of the large proportion of stems in alfalfa, they exert a strong influence on the crop’s forage quality [27]. Crude protein and crude fiber are important indexes for evaluating forage nutritional value. As plants grow, the content of digestible crude protein will gradually decrease, while the content of crude fiber will gradually increase, reducing the nutritional value of forage [28]. This is because as plants mature, the stem-to-leaf ratio increases, cell wall composition changes, and there is cell content loss, resulting in lower forage digestibility [29]. Lignin content increases with secondary cell wall thickening during plant development, and limits forage cell wall digestion by ruminants [30]. The stem acid detergent lignin (ADL) content of reduced lignin alfalfa is low, and the neutral detergent fiber digestibility (NDFD) is high [17,31]. We found that the in vitro true digestibility (IVTD) of stems in WL168 was significantly higher than that of Zhungeer, along with significantly lower NDF and ADF content. The lignin content of stems in WL168 in all developmental stages was significantly lower than that of Zhungeer (Figure 1 and Figure 2). It is well known that alfalfa exhibits low digestibility as a result of high concentrations of lignin [27], as lignin protects cellulose and hemicellulose from microbial degradation by forming a three-dimensional network structure [32]. Lignin has been reported to be the major limiting factor for forage digestibility in alfalfa [5,33]. Lignin content within alfalfa stems increases with increasing alfalfa maturation. This is because in the development process, the secondary cell wall of plants grows continuously, and the mechanical strength of plant stems is enhanced to keep plants growing upright [34]. Thus, the higher stem IVTD in WL168 compared with Zhungeer can likely be attributed to the low lignin concentration. This finding is consistent with those of Getachew et al. [22] and Grev et al. [35] who reported that reduced-lignin alfalfa has the potential to increase the plant’s forage digestibility. 

A majority of studies on solid stems and hollow stems of plants have focused on wheat [36]. Compared with ordinary wheat, solid-stem wheat has higher stem strength [37]. Though changing the quantity or composition of lignin in alfalfa by genetic manipulation, whole plant and stem digestibility can be improved [5,22,35]; however, forage nutritive value changes within stem and leaf fractions still have to be evaluated. Recently, changes in morphological development and forage nutritive value within stem and leaf fractions in reduced-lignin alfalfa have been reported, confirming that alfalfa forage quality is negatively affected by plant maturity, particularly within the stem portion of the plant [31]. In our study, we found significant differences in the anatomical structure of stems between the alfalfa cultivars (Figure 3). Interestingly, WL168 showed a unique intracellular space at stages 3 and 4, while Zhungeer did not. In addition, the length and width of the xylem in Zhungeer were greater than in WL168 at each developmental stage. The difference in lignin content between two alfalfa cultivars was verified by analyzing the anatomical structure. The results reflect those of Grev et al., who reported that any changes in forage nutritive value between alfalfa cultivars with different lignin content are likely a direct result of changes in quality occurring within the plant [31]. Hence, it would be interesting to explore the molecular mechanism behind the manipulation of stem lignification within alfalfa to enable this characteristic’s better utilization in quality improvement.

Based on the transcriptome database, 56,054 unigenes were uncovered, with an average length of 1148.69 bp and an N50 length of 1668 bp; 23,008 unigenes (41.05%) were longer than 1000 bp (Table 1). The unigenes obtained in the present study were basically in accordance with those reported for alfalfa [38,39]. These results confirmed the high quality of the transcriptome assembly. A total of 1993 genes were differentially expressed in the stems of the two alfalfa cultivars. The KEGG pathways analysis indicated that the starch and sucrose metabolism and the phenylpropanoid biosynthesis pathways were the most significantly enriched (Figure 5). Starch and sucrose metabolism plays an important role in early development, accumulating energy for subsequent growth and promoting growth [40]. The phenylpropanoid biosynthesis revealed 11 key enzymes related to lignin biosynthesis, namely *PAL*, *4CL*, *C4H*, *C3H*, *HCT*, *CCR*, *COMT*, *CCoAOMT*, *F5H*, *CAD* and *POD* [41]. These key enzymes play important roles in lignin synthesis in alfalfa stems. The identification of these lignin synthesis genes will promote the development of low-lignin alfalfa cultivars. As reported, a low-lignin alfalfa cultivar, KK179, characterized by its down-regulation of *CCoAOMT*, has been approved in many countries [42]. Here, a total of 12 differentially expressed genes, including *PAL*, *4CL*, *HCT*, *CAD*, *COMT* and *POD*, were identified in four developmental stages between the cultivars. *PAL* is at the beginning of the phenylpropionic acid pathway and one of the rate-limiting steps in lignin formation [43]. Conversion of p-cinnamic acid, caffeic acid and ferulic acid to corresponding CoA thioester was catalyzed by *4CL* [44]. *HCT* is the key enzyme for the conversion of p-coumaric acyl coenzyme A to caffeoyl coenzyme A [45]. Previous studies have shown that downregulation of *HCT* can alter both the lignin content and the monomer ratio [46]. *CAD* catalyzes three corresponding cinnamaldehydes to cinnamyl alcohol by redox reaction [47]. *CAD* may change the structure of lignin but has little effect on the total lignin content [48]. *COMT* is involved in the methylation of sinapineol and synthesis of terpineol [49]. *POD* catalyzes the polymerization of coumarin, terpineol and mustard alcohol to produce macromolecular lignin [50]. Among the 12 differentially expressed genes identified, 8 genes encoding *PAL*, *HCT*, *COMT* and *POD* showed an upward trend during the development of both cultivars. This provides an explanation for the increase of lignin content during alfalfa development. Moreover, the expression levels of 10 genes (including *PAL*, *4CL*, *HCT*, *COMT* and *POD*) involved lignin biosynthesis in Zhungeer were higher than those in WL168 except for 2 genes encoding *CAD* (Figure 6), suggesting that these genes might affect alfalfa stem lignin synthesis and further influence alfalfa quality.

## 4. Materials and Methods

### 4.1. Plant Materials and Sample Collection

Two alfalfa cultivars were employed in this study: Zhungeer and WL168. The alfalfa seeds were sterilized and planted in the experimental field of Inner Mongolia Agricultural University. The climate of the experimental sites conforms with the temperate continental monsoon climate zone, with 1600 h of sunshine per year, and annual precipitation of 330–530 mm, concentrated in July to August. The daily temperature difference is large. The annual frost-free period is 110~130 days. Base fertilizer was applied once before sowing and water was provided as needed. The sowing time was 16 May 2019. Plots consisted of 40 plants per row with a 20 cm spacing within the rows and 60 cm between rows. The field layout was a randomized complete block design with three replications of each alfalfa cultivar. Management of fertilizer and water conditions was consistent. The alfalfa plants were cut 23 May 2020 to a 4–6 cm stubble height. The sampling protocol was performed as described by Jung with modifications [3]. In brief, the seventh internode of shoots—counting from the base, initiating from the remaining nodes on previously cut stems across a range of maturity stages—were collected at four stages of maturity. The harvest dates were 9, 16 and 24 June and 16 August 2020. These sampling dates represented 17, 24, 32 and 85d of regrowth after the alfalfa plants were initially cut on 23 May 2020. Stage 1 was collected when internode 7 was just visible on the majority of new shoots. Stage 2 was taken when internode 7 was approximately half the length of internode 5 on the same shoot. Stage 3 occurred when internode 7 reached the same length as internode 5. Stage 4 was taken 53 d after the third sampling date and when internode 7 was fully mature. Three biological replicates were conducted at each stage. The stem samples used for transcriptome sequencing analysis were immediately frozen in liquid nitrogen and stored at −80 °C.

### 4.2. Nutritional Quality Analysis

NDF, ADF, IVTD and hemicellulose of both alfalfa cultivars stems at initial flowering stage were determined. The NDF and ADF were determined using the ANKOM filter bag system, as described by Vogel et al. [51]. Hemicellulose content was calculated as the difference between NDF and ADF. The 48 h IVTD were determined using in vitro techniques. Briefly, samples were first ground to pass through a 1mm screen and weighed into filter bags pre-rinsed in acetone and dried prior to filling, then incubated under anaerobic conditions in buffer solution and rumen fluid for 48 h at 39 °C. The second stage of the IVTD procedure used an NDF extraction to remove bacterial residues and other pepsin-insoluble material. The remaining product was considered undigested NDF [52].

### 4.3. Total Lignin Content Assay

Total lignin was quantified by the acetyl bromide method [25,53]. In brief, the sample was dried and ground into powder. Five mg of powder was placed into a centrifuge tube containing 0.5 mL of 25% acetyl bromide (*v*/*v* in acetic acid) and incubated at 70 °C for 30 min. After complete digestion, the sample was quickly cooled in an ice bath and mixed with 10 mL of 2M NaOH, 0.5 mL of 7.5M hydroxylamine–HCl and glacial acetic acid that was sufficient to completely dissolve the lignin extract. The standard curve was generated with alkali lignin and the absorbance was measured at 280 nm [54].

### 4.4. Histological Analysis

The collected samples were fixed in formalin acetic acid and then dehydrated successively in 70, 80, 90, 95 and 100% (*v*/*v*) ethanol. After clearing in xylene, the stem fragments were embedded in paraffin. A microtome was used to obtain 10 μm paraffin sections attached to glass slides. The paraffin sections were dried at 42 °C, stained with safranin-fast green, sealed with neutral gum, and then observed and photographed using a microscope (ZEISS SIP No. MIC01191, Oberkochen, Germany) [55].

### 4.5. cDNA Library Construction and Transcriptome Sequencing 

Total RNA of each sample was isolated using an RNA reagent kit (DP441, Tiangen Biotech, Beijing, China) following the manufacturer’s protocol. The quality and quantity of total RNA were assessed using NanoDrop 2000 analysis and gel electrophoresis. The mRNA was isolated and enriched using poly T oligo-attached magnetic beads, then lysed into short fragments using fragmentation buffer. First-strand cDNA was synthesized using reverse transcriptase and random primers. The first-strand cDNA was synthesized as a template, followed by second-strand cDNA synthesis using DNA Polymerase I and RNase H. A short cDNA fragment was obtained for PCR amplification to construct the cDNA libraries, and sequenced with an Illumina Hi-seq 4000 platform at Biomarker Technologies Co., Ltd. (Beijing, China) [56,57]. The raw sequence reads were deposited in the NCBI Sequence Read Archive (https://www.ncbi.nlm.nih.gov/sra/PRJNA628807) (accessed on 1 April 2022).

### 4.6. Transcriptome Assembly

The transcription profiles of stems of the two alfalfa cultivars in four developmental stages were explored using the Hi-seq 4000 sequencing. Three independent biological replicates were used for each treatment, resulting in 24 samples. High-quality clean reads were obtained by removing adaptor sequences along with low-quality reads (reads with a base quality less than 20). The clean reads were assembled into transcripts with Trinity (https://github.com/trinityrnaseq/trinityrnaseq) (accessed on 1 April 2022) software and, finally, used to generate unigenes [58]. The sequence assembly quality was evaluated using the number of sequences and unigenes, GC content, Q30 percentage, distribution of unigene length, mean length, and N50 length [59].

### 4.7. Annotation and Classification

Functional annotations were conducted by comparing unigene sequences with public databases, including Clusters of Orthologous Groups of proteins (COG, http://www.ncbi.nlm.nih.gov/COG/), (accessed on 1 April 2022) Gene Ontology (GO, http://www.geneontology.org), (accessed on 1 April 2022) Kyoto Encyclopedia of Genes and Genomes Plant Database (KEGG Plant Database, https://www.genome.jp/kegg/genome/plant.html), (accessed on 1 April 2022) Clusters of orthologous groups for eukaryotic complete genomes (KOG, http://www.ncbi.nlm.nih.gov/COG/KOG/kyva), (accessed on 1 April 2022) Protein family (Pfam, http://pfam.xfam.org/), (accessed on 1 April 2022) Swiss-Prot database (http://web.expasy.org/docs/swiss-prot_guideline.html), (accessed on 1 April 2022) Evolutionary genealogy of genes: Non-supervised Orthologous (eggNOG, http://eggnog.embl.de/), (accessed on 1 April 2022) NCBI non-redundant database (NR, ftp://ftp.ncbi.nlm.nih.gov/blast/db/) (accessed on 1 April 2022).

### 4.8. Differentially Expressed Genes Analysis

Differential expressed gene analysis of the samples was performed using DEseq 2 software on the BMKCloud platform (www.biocloud.net) (accessed on 1 April 2022). An FC (fold change) ≥ 2 and an FDR (false discovery rate) < 0.01 was set as the threshold. Functional annotations of the differential expressed genes were conducted by comparing unigenes sequences with GO databases. In addition, a gene set enrichment analysis of the differentially expressed genes was performed for the KEGG annotations to determine over-represented functional pathways (with a *p*-value < 0.05) at each comparison level (for different genotypes and development stages) [60,61].

### 4.9. Statistical Analysis

One-way analysis of variance (ANOVA) was used to assess differences in nutritional quality, lignin content and gene expression levels among the analyzed stem tissues of alfalfa using SPSS 21.0 software. Differences were considered as significant at *p* < 0.05. Graphs were drawn with Sigmaplot 14.0. All experiments included at least three biological replicates.

## 5. Conclusions

In this study, two alfalfa cultivars were used to explore stem development. The results indicated that the IVTD of WL168 stems was significantly higher than that of Zhungeer stems, along with significantly lower NDF, ADF and lignin content. Moreover, intracellular space appeared in the stems of WL168 from stage 3, while in Zhungeer this did not occur. Furthermore, most differentially expressed genes involved in lignin synthesis, including *PAL*, *4CL*, *HCT*, *COMT* and *POD* in Zhungeer, were higher than in WL168 except for 2 genes encoding *CAD*. These results revealed different stem digestibility and structure in alfalfa cultivars may be related to the expression of lignin synthesis genes. Profiling the expression of genes involved in lignin biosynthesis-related enzymes and understanding lignin distribution and content in stem may help elucidate the regulatory mechanisms of lignin biosynthesis alfalfa. This may provide a reference for screening alfalfa germplasms of low lignin or high forage quality utilizing the morphological or molecular markers.

## Figures and Tables

**Figure 1 plants-11-02601-f001:**
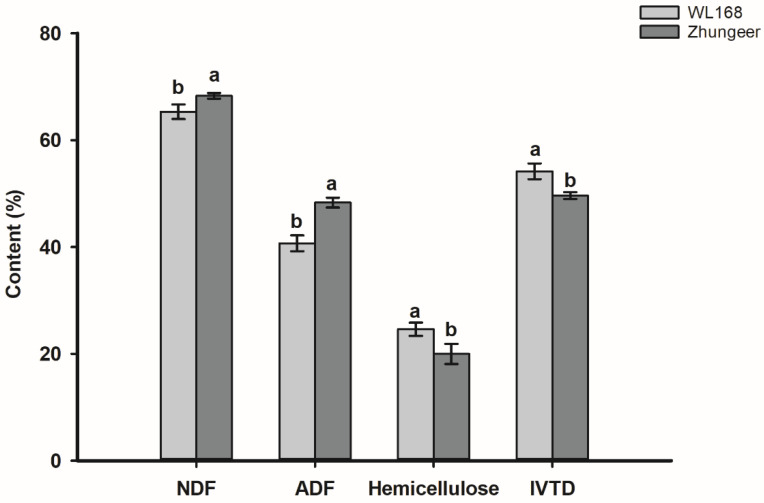
Nutritional quality analysis of stems at initial flowering stage of two alfalfa cultivars. NDF, neutral detergent fiber; ADF, acid detergent fiber; IVTD, in vitro true digestibility. Different letters indicate significant differences at *p* < 0.05.

**Figure 2 plants-11-02601-f002:**
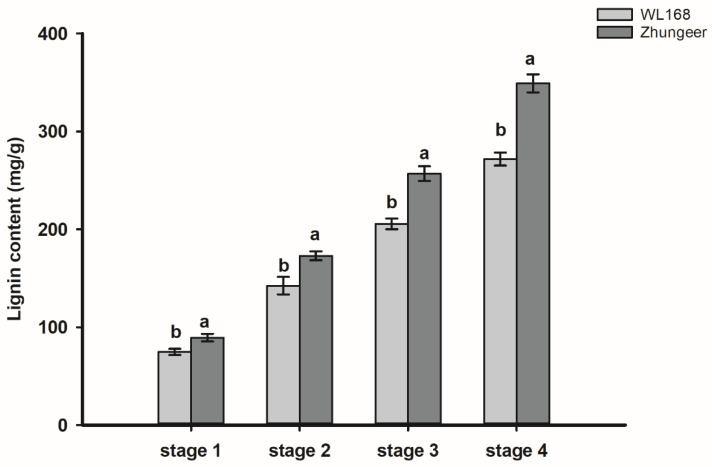
Lignin content of stems of two alfalfa cultivars in four developmental stages. Different letters indicate significant differences at *p* < 0.05.

**Figure 3 plants-11-02601-f003:**
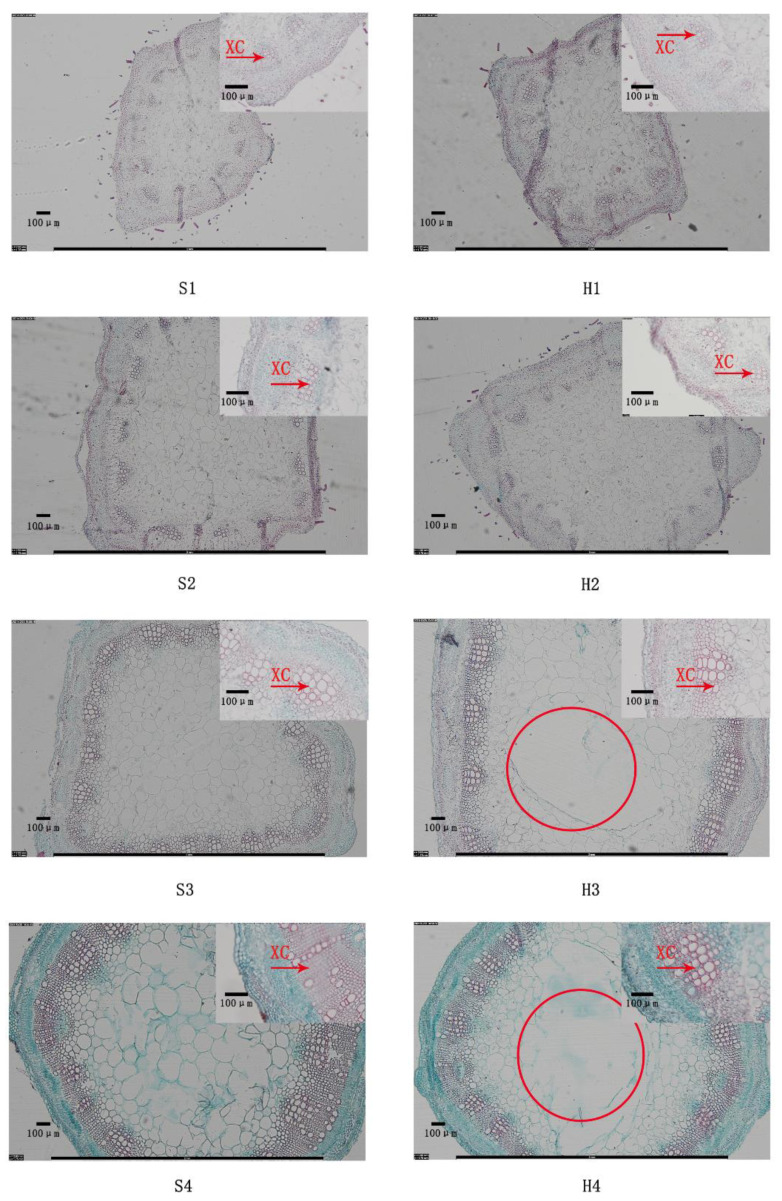
Transverse sections at stems of two alfalfa cultivars in four developmental stages. XC, xylem cell. The red circle represents intracellular space. S1–S4 represent four stages of development in Zhungeer. H1–H4 represent four stages of development in WL168.

**Figure 4 plants-11-02601-f004:**
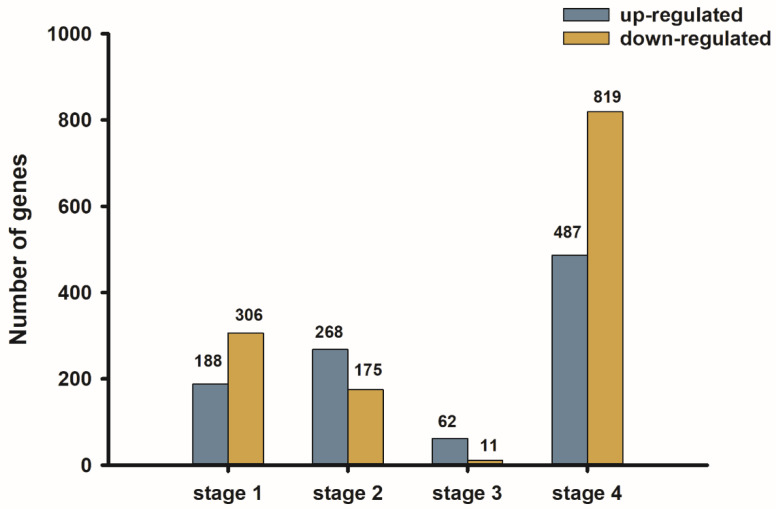
The number of differentially expressed genes in four development stages.

**Figure 5 plants-11-02601-f005:**
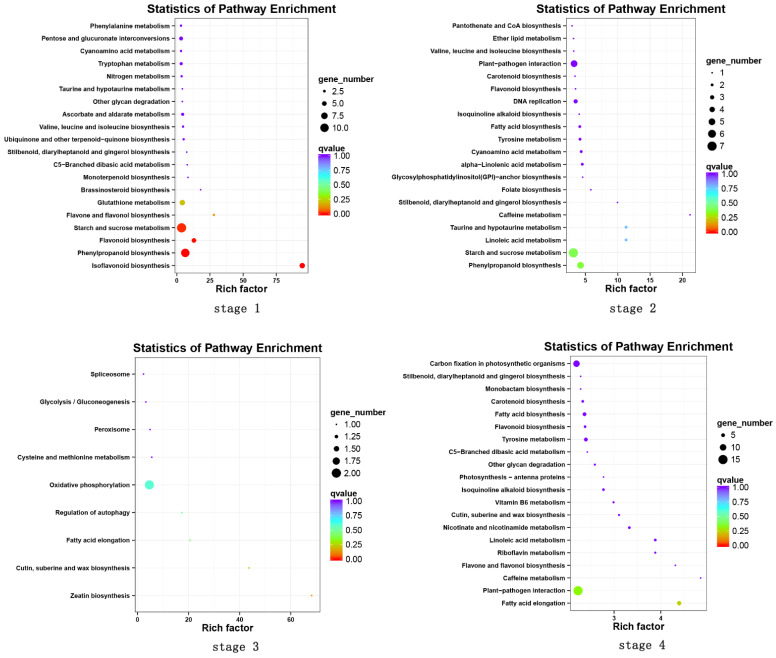
Scatter plot of KEGG pathways of the differentially expressed genes enriched.

**Figure 6 plants-11-02601-f006:**
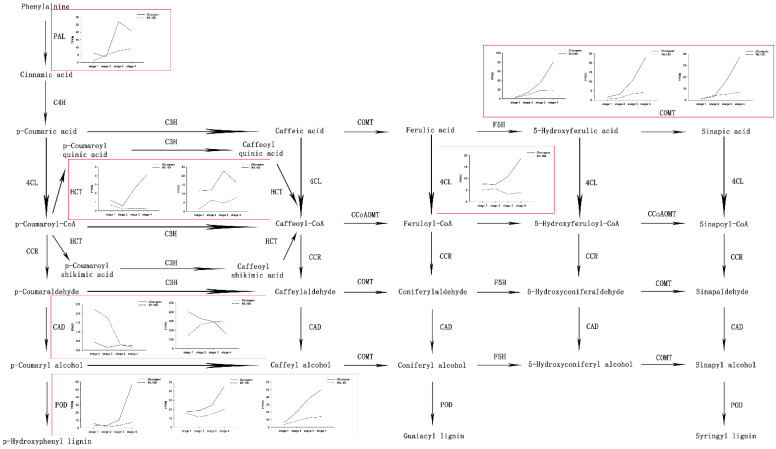
Lignin biosynthetic pathways. FPKM, fragments per kilobase of exon model per million mapped fragments. The real line represents Zhungeer and the imaginary line represents WL168.

**Table 1 plants-11-02601-t001:** Statistics of assembly results.

Length Range	Transcript	Unigene
300–500	26,086 (20.85%)	16,511 (29.46%)
500–1000	36,879 (29.48%)	16,535 (29.50%)
1000–2000	39,075 (31.24%)	14,339 (25.58%)
>2000	23,053 (18.43%)	8669 (15.47%)
Total Number	125,093	56,054
Total Length	161,338,984	64,388,875
N50 Length	1761	1668
Mean Length	1290	1148.69

**Table 2 plants-11-02601-t002:** Functional annotation of unigenes.

Database	Number Annotated	Percentage (%)
COG	15,281	27.26
GO	25,913	46.23
KEGG	17,045	30.41
KOG	23,860	42.57
Pfam	30,672	54.72
Swissprot	25,608	45.68
eggNOG	40,075	71.49
Nr	44,610	79.58
Overall	45,398	80.99

**Table 3 plants-11-02601-t003:** GO analysis for differentially expressed genes of two alfalfa cultivars in four developmental stages.

GO Terms	H1 vs. S1	H2 vs. S2	H3 vs. S3	H4 vs. S4
**Biological Process**				
metabolic process	137	95	10	390
cellular process	115	93	11	398
single-organism process	101	75	10	272
biological regulation	34	38	2	137
localization	16	13	1	76
response to stimulus	42	33	4	109
cellular component organization or biogenesis	12	21	1	74
signaling	14	16	2	51
developmental process	4	11	2	24
multicellular organismal process	11	11	2	24
reproductive process	8	3	0	11
multi-organism process	15	3	0	5
**Cellular Component**				
cell	71	61	6	322
cell part	71	61	6	321
membrane	101	68	9	242
organelle	34	40	4	222
membrane part	86	54	8	207
macromolecular complex	7	14	1	78
organelle part	15	18	2	109
membrane-enclosed lumen	1	4	0	25
extracellular region	13	7	0	8
cell junction	9	4	0	7
symplast	9	4	0	7
**Molecular Function**				
catalytic activity	142	101	11	362
binding	110	94	13	396
transporter activity	8	5	1	42
structural molecule activity	0	2	0	18
nucleic acid binding transcription factor activity	4	1	0	9
molecular transducer activity	3	2	1	5
signal transducer activity	3	2	1	5
antioxidant activity	8	3	0	3
electron carrier activity	0	1	1	5
transcription factor activity, protein binding	2	0	0	4
nutrient reservoir activity	2	0	0	0

## Data Availability

The datasets presented in this study can be found in online repositories. The names of the repository/repositories and accession number(s) can be found at: https://www.ncbi.nlm.nih.gov/ (accessed on 1 April 2022), PRJNA628807.

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
