# Peer review of "Comparative Transcriptome and Anatomic Characteristics of Stems in Two Alfalfa Genotypes"

_plants, 2022, doi:10.3390/plants11192601_

Round 1
Reviewer 1 Report
The topic is relevant for Plants, however the study needs many improvements. All detailed comments and suggestions can be found in the attached file.

Reviewer 2 Report
I checked your manuscript and described comments below.
Alfalfa is a plant that is used as livestock feed worldwide. In some areas it is also edible.
This paper does a very good transcriptome analysis of Alfalfa.
I described following problem.
1. The characters in Figure5 and Figure6 are too small. I can't comment, so I think it's better to fix it.
2. KEGG has different databases depending on species. You should use KEGG Plant. I think it's better to check and write.
I don't think this paper has any major mistakes or grammatical problems.
Round 2
Reviewer 1 Report
The authors did a appreciable work! Congratulations!